# *A Good Learner can Teach Better*: TEACHER-STUDENT COLLABORATIVE KNOWLEDGE DISTILLATION

**Ayan Sengupta**[1], **Shantanu Dixit**[2], **Md Shad Akhtar**[2], **Tanmoy Chakraborty**[1]
[1]Indian Institute of Technology Delhi, India
[2]Indraprastha Institute of Information Technology Delhi, India

`ayan.sengupta@ee.iitd.ac.in`,`{shantanu20118, shad.akhtar}@iiitd.ac.in`,
`tanchak@iitd.ac.in`

## ABSTRACT

Knowledge distillation (KD) is a technique used to transfer knowledge from a larger "teacher" model into a smaller "student" model. Recent advancements in meta-learning-based knowledge distillation (MetaKD) emphasize that the fine-tuning of teacher models should be aware of the student's need to achieve better knowledge distillation. However, existing MetaKD methods often lack incentives for the teacher model to improve itself. In this study, we introduce `MPDistil`, a meta-policy distillation technique, that utilizes novel optimization strategies to foster both *collaboration* and *competition* during the fine-tuning of the teacher model in the meta-learning step. Additionally, we propose a curriculum learning framework for the student model in a competitive setup, in which the student model aims to outperform the teacher model by self-training on various tasks. Exhaustive experiments on SuperGLUE and GLUE benchmarks demonstrate the efficacy of `MPDistil` compared to 20 conventional KD and advanced MetaKD baselines, showing significant performance enhancements in the student model – e.g., a distilled 6-layer BERT model outperforms a 12-layer BERT model on five out of six SuperGLUE tasks. Furthermore, `MPDistil`, while applied to a large language teacher model (DeBERTa-v2-xxlarge), significantly narrows the performance gap of its smaller student counterpart (DeBERTa-12) by just $4.6\%$ on SuperGLUE. We further demonstrate how higher rewards and customized training curricula strengthen the student model and enhance generalizability.

## 1 INTRODUCTION

Large language models (LLMs) like GPT-3 (Brown et al., 2020), PaLM (Chowdhery et al., 2022), and LLaMA (Touvron et al., 2023) have demonstrated remarkable performance in a wide range of natural language tasks, showcasing their abilities in tasks requiring minimal, zero, or context-based prior knowledge (Kossen et al., 2023). These models underwent extensive pre-training on massive datasets to achieve such impressive capabilities, incurring substantial computational costs, as highlighted by Zhu et al. (2023). The scaling laws associated with language models (Kaplan et al., 2020) indicate that achieving superior predictive performance comes at the cost of a large number of model parameters, extensive training data, and substantial computational power. However, these costs typically do not account for inference expenses, which must be handled by resource-constrained devices when deploying these models at scale. One approach to address this challenge is "model compression" (Deng et al., 2020), a technique designed to reduce computational requirements and storage while preserving the original models' performance.

Knowledge distillation (KD) (Hinton et al., 2015) is a widely adopted model compression method that entails transferring knowledge from a larger and more complex model, referred to as the "teacher model", to a smaller and simpler model known as the "student model". In traditional white-box knowledge distillation, the primary focus is on minimizing the "teacher-student margin" by narrowing the gap between the teacher's distribution, $t_\theta(\boldsymbol{y}|\boldsymbol{x})$, and the student's distribution, $s_\theta(\boldsymbol{y}|\boldsymbol{x})$. More advanced knowledge distillation techniques (Sun et al., 2019; Jiao et al., 2019) go further by aligning the intermediate feature maps between the teacher and student models. This alignment helps the student replicate the behaviors demonstrated by the teacher model. However, most of these techniques operate in a unidirectional manner, in which the teacher model remains fixed after pre-training or

fine-tuning, while the student model is trained to minimise the teacher-student margin. Zhou et al. (2021) pointed out several limitations of this approach. Existing approaches do not optimize the teacher explicitly for the distillation task or consider the student model's capacity during training. This differs from the real-life teacher-student dynamic, in which a teacher continuously enhances her knowledge and teaching skills through repeated student interactions. To address this gap, Zhou et al. (2021) introduced MetaDistil, a meta-learning-based knowledge distillation framework that aims to overcome this limitation by considering the student's predictive performance while fine-tuning the teacher model. In MetaDistil, the teacher model undergoes training on a separate quiz dataset to minimize the knowledge distillation loss between the teacher and student models along with the teacher's task-specific loss. Although MetaDistil addresses several shortcomings of conventional KD methods, it is still far from imitating real-world teacher-student interactions. In a real-world educational context, teachers may not solely focus on narrowing the teacher-student knowledge gap to improve teaching; instead, they aim to maximize the *joint* performance of students and teachers. Furthermore, in a typical classroom setting, students study various subjects and follow a curriculum to optimize their learning across all subjects. For example, a student might aim to improve her understanding of Physics by studying selected concepts from Mathematics. However, existing knowledge distillation methods tend to distil knowledge for different tasks in isolation, which means they fail to capture the commonalities among tasks. Consequently, the distilled models struggle to generalize effectively to new tasks with limited training data. Another significant challenge with the sole optimization of the teacher-student margin is that it positions the teacher as the "sole role model" for the student without encouraging the student to surpass the teacher's performance.

To address these limitations, we introduce `MPDistil`, a meta-policy knowledge distillation framework that employs a collaborative learning approach within the context of meta-knowledge distillation[1]. Through empirical analyses, we underscore that optimizing a shared utility function can enhance the teacher's predictive capabilities and ability to impart knowledge to the student model. Furthermore, we establish a meta-reinforcement learning-based "curriculum learning" paradigm in which the student model undergoes fine-tuning on a curriculum – a sequence of tasks. The aim is to enhance the student model's generalization skills, enabling it to surpass the teacher model. This competitive approach empowers the student to outperform the teacher. We assess the effectiveness of `MPDistil` using two natural language understanding benchmarks, SuperGLUE (Wang et al., 2019) and GLUE (Wang et al., 2018), encompassing 15 NLU tasks. Following the experimental setup adopted by contemporary KD-related studies, we highlight the effectiveness of `MPDistil` with BERT-base (Devlin et al., 2018) as the teacher model and BERT 6L (with six layers) as the student model. On the SuperGLUE benchmark, the meta update enhances the BERT teacher's performance with a maximum margin of $+3\%$, resulting in a notable improvement in student performance by $+5.9\%$. Surprisingly, the distilled student surpasses the teacher model on five of six SuperGLUE tasks within this competitive framework, with a maximum advantage of $+7\%$. In contrast, the most competitive baseline, MetaDistil, exhibits an average deficit of $-1.1\%$, indicating that the distilled student falls short of the teacher model's performance. We also extend our evaluation to a large language model, DeBERTa-v2-xxlarge (He et al., 2020), as the teacher model and its smaller variant, DeBERTa-12, as the student model. Ours is the first large-scale study highlighting the effectiveness of distilling knowledge from large language models ($> 1B$ parameters) to smaller models. `MPDistil`, when applied to DeBERTa, improves the student model's performance with a maximum margin of $+2.2\%$, reducing the student-teacher margin to $-4.6\%$ (a negative student-teacher margin indicates that student underperforms the teacher), which is significantly higher than the other distillation methods ($-7.9\%$).

## 2 RELATED WORK

**Knowledge distillation.** One of the earliest investigations on knowledge distillation by Hinton et al. (2015) introduced a technique in which the student model is trained to minimize the task-specific loss while also enhancing the similarity between the outputs of the teacher and student models. This enhancement is achieved by minimizing the difference between softened class probabilities generated by both the teacher and student model using temperature scaling. Sanh et al. (2019) distilled a more compact version of BERT-base by halving the number of layers and removing the pooler output and token type embeddings. In contrast to learning solely from the teacher's

---

[1]Source code of `MPDistil` can be found at `https://github.com/notmyname16/MPDistil`.

last layer, Sun et al. (2019) introduced a "patient learner" approach. The student learns from the last $k$ layers by incorporating a Mean Squared Error (MSE) loss between the teacher and student layer representations, leveraging the internal representations of the teacher model. TinyBERT (Jiao et al., 2019) also utilises the internal representations of BERT to distil knowledge. It achieves this by distilling the transformer, embedding, and classification layers. Yang et al. (2020) and Wu & Chiu (2020) introduced multiple teacher-based knowledge distillation – each teacher is assigned predetermined weights, and their weighted average probability distributions are used to train the student model. In a similar attempt, Yuan et al. (2021) proposed a reinforcement learning-based technique in which different weights are assigned to various teacher models during distillation using a policy function.

**Meta knowledge distillation.** Meta-learning-based knowledge distillation techniques utilize a meta-learning loop to enhance the teacher model. Established MetaKD approaches (Pan et al., 2020; Liu et al., 2022b) employ meta-teachers to adapt them to new domains and tasks. Liu et al. (2022b) argued that using fixed temperature can result in suboptimal knowledge transfer from teacher to student. They aimed to meta-learn the student and teacher's temperature along with the distillation function. In contrast, Zhou et al. (2021) introduced a generalized framework, MetaDistil, focusing on teaching teachers how to facilitate knowledge distillation. Note that these techniques entail fine-tuning a meta-teacher, which is computationally equivalent to the teacher model. Fine-tuning a large pre-trained teacher model can be prohibitively expensive. Furthermore, the meta-learning objective optimized by the meta-teacher primarily aims to narrow the gap between the teacher and student models without encouraging them to surpass each other.

**Curriculum based knowledge distillation.** Curriculum learning (Bengio et al., 2009) involves increasing the level of training samples progressively during training. Xiang et al. (2020) designed curricula based on the confidence score of the mixture of expert models, which was later used to train a unified student model. Li et al. (2023) proposed learning a curriculum from easy to hard levels through a learnable temperature optimized through a reversed gradient. It involved increasing the distillation loss through a cosine schedule, leading to an increased difficulty for student learning.

Our work introduces collaborative and competitive utility optimization within the meta-knowledge-distillation paradigm and establishes a competitive reward system for the student using task based curricula. This competitive framework motivates both teacher and student models to outperform each other. The joint utility can also be optimized using a simpler feed-forward meta-teacher, eliminating the need to fine-tune the teacher backbone. This enhances the generality of our proposed method, making it suitable for distilling large pre-trained models and making our method more computationally efficient.

## 3 PROPOSED METHODOLOGY

Our proposed **m**eta-**p**olicy knowledge **distil**lation method, `MPDistil`, aims to distil knowledge from an arbitrary teacher model, parameterized by $\theta_T$ to an arbitrary student model, parameterized by $\theta_S$. Although the teacher and student models can belong to any family (e.g., encoder, encoder-decoder or decoder), we typically assume that they belong to the same architectural family. Additionally, we assume that both the teacher and student models can be extended to a multi-task framework, wherein the models can be optimized on any task $\mathcal{T} \in \mathbb{T}$. `MPDistil` consists of four steps – (i) fine-tuning the teacher model, (ii) distilling the teacher model to the student model, (iii) meta-teacher learning, and (iv) student curriculum learning. We illustrate these steps in Figure 1.

**Fine-tuning teacher model.** For any given task $\mathcal{T} \in \mathbb{T}$, we use the task-specific loss function $\mathcal{L}_{\mathcal{T}}$ to optimize and fine-tune the teacher model. For classification tasks, cross-entropy is used as the task-specific loss, whereas for regression tasks, MSE is used to calculate the loss. Precisely, for a given training data batch of size $N$, $\mathcal{D}_{train} = \{(x_1, y_1), \ldots, (x_N, y_N)\}$, we calculate $\mathcal{L}_{\mathcal{T}}^{teacher} = \frac{1}{N} \sum_{i=1}^{N} \mathcal{L}_{\mathcal{T}}(y_i, \hat{y}_{(i,T)})$, with $\hat{y}_{(i,T)} = T(x_i; \theta_T)$, the output generated by the teacher model $T$ with parameters $\theta_T$. The teacher model parameters are then updated with gradients computed on $\mathcal{L}_{\mathcal{T}}$ with respect to $\theta_T$. Although we highlight the teacher fine-tuning with vanilla training procedure, for large language models, it can be replaced with parameter-efficient fine-tuning or even few-shot or zero-shot fine-tuning.

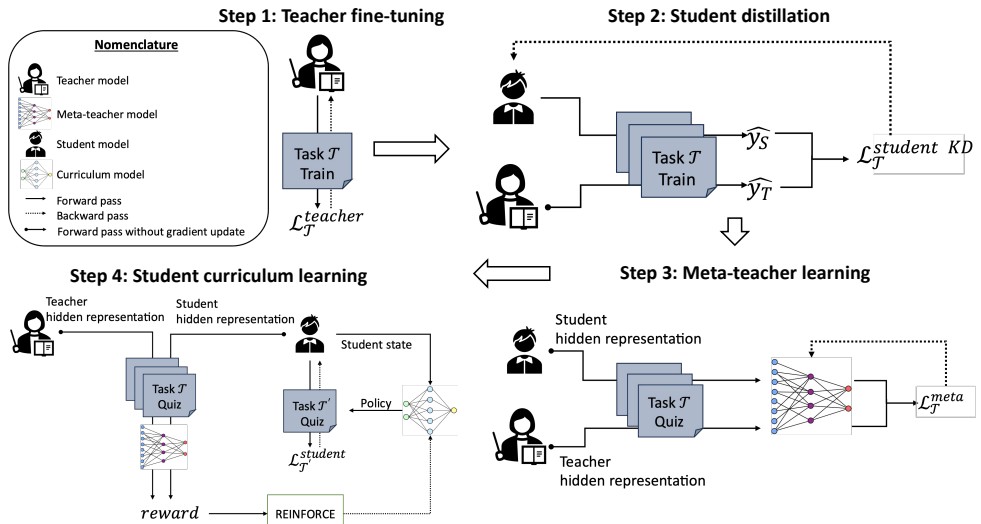

Figure 1: A schematic diagram of our proposed `MPDistil` framework. It consists of four steps – (i) *teacher fine-tuning step*, where the original teacher model is fine-tuned on task $\mathcal{T}$, (ii) *student distillation step*, where the student model is fine-tuned with task-specific and knowledge distillation loss, (iii) *meta-teacher learning step*, where a feed-forward model, the meta-teacher, is trained to jointly optimize the outputs for the teacher and student hidden representations, and (iv) in *student curriculum learning step*, we train a student curriculum model to generate a suitable curriculum, a sequence of tasks $\mathcal{T}' \in \mathbb{T}$ for which the distilled student is fine-tuned and calculated rewards.

**Distilling teacher model to student model.** In step 2 of our framework, we distil the student model $S$ with the knowledge acquired by the teacher model with fine-tuning. We use the task-specific loss $\mathcal{L}_{\mathcal{T}}$ along with a knowledge distillation loss $\mathcal{L}_{\mathcal{KD}}$ computed on the training dataset. `MPDistil` is a general framework and can handle any arbitrary $\mathcal{L}_{\mathcal{KD}}$ function, as long as it is differentiable with respect to the student model parameters $\theta_S$. Following Zhou et al. (2021), we use a convex combination of the MSE loss between the teacher and student logits and the mean squared differences between the teacher and student hidden representations. Formally, we calculate,

$$\mathcal{L}_{\mathcal{T}}^{\text{student KD}} = \frac{1}{N}\sum_{i=1}^{N}\alpha\mathcal{L}_{\mathcal{T}}(y_i,\hat{y}_{(i,S)}) + (1-\alpha)\left\|\Phi\left(\frac{\hat{y}_{(.,T)}}{\tau}\right) - \Phi\left(\frac{\hat{y}_{(.,S)}}{\tau}\right)\right\|_2 + \beta\left\|h_{(.,T)} - h_{(.,S)}\right\|_2$$
(1)

Here, $\Phi$ denotes the $Softmax$ activation function. Although some studies (Hinton et al., 2015; Gu et al., 2023) use Kullback–Leibler Divergence (KLD) and reverse KLD between the teacher and student logits, following Kim et al. (2021), we use $L2$ norm to calculate the difference between teacher and student logits, to stabilize the optimization. Hence, a student optimized with $\mathcal{L}_{\mathcal{T}}^{\text{student KD}}$ is not only trained on the task $\mathcal{T}$ standalone but also acquires the knowledge from the teacher through soft label association and shared latent, hidden representations.

**Meta-teacher learning.** In step 3 of `MPDistil`, we jointly update the teacher model parameters to optimize the teacher and student performances. To achieve this, we devise a meta-teacher $T'$, parametrized with $\theta_{T'}$. The meta-teacher follows a simple feed-forward network architecture with the ability to digest the hidden state representations from both the teacher and student models and generate the final output. In our white-box distillation framework, we assume that the output logits and hidden representations are accessible from both the teacher and student models. The meta-teacher, therefore, can act as a discriminator, separating the representations extracted by the original teacher and the student models. We train the meta-teacher parameters $\theta_{T'}$ as the meta-learner, whereas the student model parameters $\theta_S$ are learned as the inner-learner. It is interesting to notice that as opposed to the meta-teacher designed by Zhou et al. (2021), our meta-teacher has only an insignificant number (only $0.001\%$ of the teacher model) of learnable parameters and, therefore,

is much easier to train in the meta-learning setup. We avoid a large pre-trained teacher model as cloning it to a meta-teacher and fine-tuning could be prohibitively expensive. Our meta-teacher is, therefore, computationally more efficient, even for distilling from computationally large pre-trained models.

We adopt a quiz dataset $\mathcal{Q}_{\mathcal{T}}$ for each task $\mathcal{T}$, separate from the original training dataset for meta-teacher learning. We formulate two different training objectives for the meta-teacher. In the first approach, we compute a *collaborative loss*, in which the meta-teacher optimizes the joint task-specific losses calculated with the teacher's and student's hidden representations. Given a training data $(x_i, y_i)$, we obtain the hidden states from the teacher and student models as $h_{(i,T)}$ and $h_{(i,S)}$, respectively, and obtain the final output from the meta-teacher model as, $\hat{\hat{y}}_{(i,T)} = T'(h_{(i,T)}; \theta_{T'})$ and $\hat{\hat{y}}_{(i,S)} = T'(h_{(i,S)}; \theta_{T'})$, respectively. If the task $\mathcal{T}$ is a classification task, we use only the class probabilities of the true class $\bar{y}_{(i,T)} = \Phi(\hat{\hat{y}}_{(i,T)})_c$ and $\bar{y}_{(i,S)} = \Phi(\hat{\hat{y}}_{(i,S)})_c$ for $c = \arg\max_k y_{(i,k)}$, the true class label. The collaborative loss is formally defined as,

$$
\mathcal{L}_{\mathcal{T}}^{\text{meta col}} = \begin{cases} -\dfrac{1}{2N} \sum_{i=1}^{N} \Big[ \log \bar{y}_{(i,T)} + \log \bar{y}_{(i,S)} \Big], & \text{if } \mathcal{T} \text{ is a classification task} \\[2ex] \dfrac{1}{2} \Big\| y - \hat{\hat{y}}_{(.,T)} \Big\|_2 + \dfrac{1}{2} \Big\| y - \hat{\hat{y}}_{(.,S)} \Big\|_2, & \text{if } \mathcal{T} \text{ is a regression task} \end{cases}
\tag{2}
$$

The meta-teacher acts as a discriminator and does not understand whether the teacher or the student obtains the hidden representation. Moreover, the original student and teacher parameters are frozen during meta-teacher training, and the meta-teacher is trained. Therefore, the meta-teacher only learns the optimal function to jointly optimize the teacher's and student's final predictive performances. We compute a *competitive loss* in another formulation between the teacher and student representations. The competitive loss is motivated by the Wasserstein loss used in WGAN (Arjovsky et al., 2017) and calculates the Earth Mover's (EM) distance between the teacher and student logits. Additionally, we add the meta-teacher's task-specific loss so that the meta-teacher can minimize its loss, along with maximizing the margin from the student model. Using the Kantorovich-Rubenstein duality (Villani et al., 2009), we compute the competitive loss as,

$$
\mathcal{L}_{\mathcal{T}}^{\text{meta com}} = \begin{cases} -\dfrac{1}{N} \sum_{i=1}^{N} \Big[ 2 \log \bar{y}_{(i,T)} - \log \bar{y}_{(i,S)} \Big], & \text{if } \mathcal{T} \text{ is a classification task} \\[2ex] \Big\| y - \hat{\hat{y}}_{(.,T)} \Big\|_2 - \dfrac{1}{2} \Big\| y - \hat{\hat{y}}_{(.,S)} \Big\|_2, & \text{if } \mathcal{T} \text{ is a regression task} \end{cases}
\tag{3}
$$

**Proposition 1.** For any classification task $\mathcal{T}$, having $\mathbb{E}[\mathcal{L}_{\mathcal{T}}^{\text{meta col}}] < \mathbb{E}[\mathcal{L}_{\mathcal{T}}^{\text{meta com}}]$ ensures stronger student with $\mathbb{E}[\bar{y}_S] > \mathbb{E}[\bar{y}_T]$.[2]

**Student curriculum learning.** The final step of MPDistil is the student curriculum learning, in which the student model aims to outperform the meta-teacher by training itself on a suitable set of tasks, possibly different from the task at hand $\mathcal{T}$. Curriculum learning (Bengio et al., 2009) involves gradually acquiring skills by initially focusing on simpler tasks before progressing to more complex ones. We use a policy learner, a curriculum model, to learn the curriculum for the student. The curriculum model is a feed-forward network that takes the current state of the student model and samples a task (action) $\mathcal{T}' \in \mathbb{T}$ in each forward pass. The student model is trained on a batch sampled from $\mathcal{Q}_{\mathcal{T}'}$ and is used to calculate a reward on $\mathcal{Q}_{\mathcal{T}}$. For an updated student model, we calculate the output $\hat{y}_{(i,S)} = S(x_i; \theta_S)$ for each $(x_i, y_i) \sim \mathcal{Q}_{\mathcal{T}}$. Similarly, we obtain the output $\hat{y}_{(i,T')}$ from the meta-teacher. Analogous to the previous meta-teacher learning step, we obtain only the predicted probability of the true output class for classification tasks. We evaluate two different reward functions: (i) binary reward and (ii) real reward.

$$
R^{\text{binary}} = \begin{cases} \mathbb{I}_{\hat{y}_{(i,S)} > \hat{y}_{(i,T')}}, & \text{if } \mathcal{T} \text{ is a classification task} \\[2ex] \mathbb{I}_{\left\| y_i - \hat{y}_{(i,T')} \right\|_2 > \left\| y_i - \hat{y}_{(i,S)} \right\|_2}, & \text{if } \mathcal{T} \text{ is a regression task} \end{cases}
\tag{4}
$$

---

[2]All the proofs presented in the paper are supplied in Appendix A.

Here, $\mathbb{I}$ is the indicator function. Similarly,

$$R^{\text{real}} = \begin{cases} \hat{y}_{(i,S)} - \hat{y}_{(i,T')}, & \text{if } \mathcal{T} \text{ is a classification task} \\ \left\| y_i - \hat{y}_{(i,T')} \right\|_2 - \left\| y_i - \hat{y}_{(i,S)} \right\|_2, & \text{if } \mathcal{T} \text{ is a regression task} \end{cases} \qquad (5)$$

**Proposition 2.** For any classification task $\mathcal{T}$, $\mathbb{E}[R^{\text{binary}}] \geq \mathbb{E}[R^{\text{real}}]$.

The expected reward for an entire episode is calculated as $G_t = \sum_{k=t+1}^{E} \gamma^{k-t-1} R_k$, with a suitable discount factor $\gamma$ and the length of the episode $E$. In our case, the length of an episode is the number of quiz batches for task $\mathcal{T}$. We use the Monte Carlo policy gradient algorithm, RE-INFORCE (Williams, 1992; Sutton et al., 1999), to update the curriculum model parameters $\theta_A$ to maximize the expected reward. The curriculum model samples tasks that maximize the reward for the student model, i.e., increases the difference between the student and teacher models. In other words, having trained on these newly sampled tasks, the student model can achieve better results than the teacher when evaluated through meta-teacher, leading to higher rewards.

## 4 Experimental Details and Results

Following the contemporary knowledge distillation studies (Liu et al., 2022a; Zhou et al., 2021), we evaluate `MPDistil` on 15 different natural language understanding tasks from SuperGLUE (Wang et al., 2019) and GLUE (Wang et al., 2018) benchmarks. We choose the classification tasks – CB, COPA, RTE, WiC, WSC and BoolQ from the SuperGLUE benchmark. From the GLUE benchmark, we use MNLI, MNLI Mismatch, QNLI, QQP, WNLI, MRPC, RTE, SST-2 and STS-B tasks, out of which only STS-B is a regression task and the remaining are classification tasks. We elaborate on these tasks in Appendix B.1. We utilize the pre-trained BERT-base (Devlin et al., 2018) model as the teacher ($111M$ parameters) and the pre-trained BERT 6-layer model as the student ($66M$ parameters)[3]. Additionally, we evaluate `MPDistil` on a pre-trained large language model, DeBERTa-v2-xxlarge (He et al., 2020) ($1.4B$ parameters) and its 12-layered variant ($547M$ parameters) as the student model. As described in Section 3, for teacher, meta-teacher and student models, we use multiple output heads depending on the number of tasks. For SuperGLUE and GLUE, we use six and nine output heads (one for each task), respectively. However, all the models are trained on each task separately. The BERT-base model utilizes a pooling layer for learning a suitable projection from the output representation corresponding to the first token ([CLS] token) and uses it for downstream tasks. We use the student model pooling layer weight matrix and learn a projection vector on $\mathbb{R}^{768}$ to represent the state of the student model in curriculum learning. Meta-teacher learning and student curriculum learning require a separate labelled dataset, for which we split the original training dataset provided in SuperGLUE and GLUE tasks into an updated training and quiz sets by $9:1$. However, we use the original datasets provided in the benchmarks for dev and test. We furnish other experimental details in Appendix B.2. We evaluate `MPDistil` with 20 contemporary KD and MetaKD methods, some of which are elaborated in Appendix B.3.

**Results.** We report the performances of different models – the fine-tuned teacher model, the distilled student, the meta-teacher and the updated student with `MPDistil` on the SuperGLUE tasks in Table 1. Additionally, we report the results obtained from the student model when fine-tuned on the tasks without distillation. As an ablation to `MPDistil`, we evaluate the performance of an alternate student model that is only trained with $\mathcal{L}_{KD}$ in step 4 (c.f. Figure 1), without curriculum learning. We denote this model as '(-) Curriculum learning'. On the SuperGLUE dev split, the meta-teacher obtains $0.7\%$ better performance than the original teacher model. Although both the loss formulations are adequate for the meta-teacher, collaborative loss helps the meta-teacher obtain better performance, as opposed to the competitive loss. Collaborative loss is equally essential for better student distillation. On SuperGLUE dev, the student distilled with collaborative loss and binary reward achieves $5.9\%$ and $1.5\%$ better than the base student and teacher models, respectively. Although both reward functions are equally effective, the distilled student achieves $0.2\%$ improvement with binary reward than with the real reward function. Without student curriculum learning, the ablation model achieves $2.4\%$ better than the base student model. However, the improvement over the teacher model remains muted at $-2.1\%$, indicating the positive impact of curriculum learning on the student.

[3]The pre-trained models are obtained from `https://huggingface.co/models/`.

| Methods | BoolQ | CB | COPA | RTE | WiC | WSC |
|---|---|---|---|---|---|---|
| Teacher (BERT-base) | 75.3 | 83.9 | 63.0 | 67.1 | 57.1 | 64.4 |
| Student (BERT-base 6L) | 71.6 | 75.0 | 53.0 | 64.6 | 56.0 | 63.5 |
| Distilled Student | 73.0 | 80.4 | 54.0 | 66.4 | 57.7 | 64.4 |
| Meta-teacher | | | | | | |
| (+) Col loss | 75.5 | 83.9 | 66.0 | 67.8 | 58.5 | 63.5 |
| (+) Com loss | 75.5 | 83.9 | 63.0 | 67.8 | 57.2 | 61.5 |
| Student with MPDistil | | | | | | |
| (+) Col loss + Binary reward | 72.8 | 83.9 | 67.0 | 67.1 | 58.0 | 65.4 |
| (+) Col loss + Real reward | 73.4 | 82.1 | 70.0 | 66.4 | 58.6 | 64.4 |
| (+) Com loss + Binary reward | 73.0 | 80.4 | 62.0 | 67.5 | 59.6 | 65.4 |
| (+) Com loss + Real reward | 73.0 | 78.6 | 63.0 | 66.4 | 58.9 | 65.4 |
| (-) Curriculum learning + Col loss | 72.5 | 78.6 | 58.0 | 65.3 | 58.3 | 64.4 |
| (-) Curriculum learning + Comp loss | 72.3 | 76.8 | 59.0 | 65.3 | 58.0 | 63.5 |

(a) BERT Teacher

| Methods | BoolQ | CB | COPA | RTE | WiC | WSC |
|---|---|---|---|---|---|---|
| Teacher (DeBERTa-v2-xxlarge) | 86.8 | 82.1 | 84.0 | 90.6 | 57.7 | 63.4 |
| Student (DeBERTa-v2 12L) | 82.1 | 82.1 | 64.0 | 79.4 | 57.8 | 63.5 |
| Distilled Student | 82.4 | 76.9 | 67.0 | 78.7 | 59.4 | 64.4 |
| Meta-teacher | | | | | | |
| (+) Col loss | 86.6 | 76.9 | 83.0 | 89.5 | 55.9 | 63.5 |
| (+) Com loss | 86.9 | 78.5 | 83.0 | 89.5 | 56.9 | 63.5 |
| Student with MPDistil | | | | | | |
| (+) Col loss + Binary reward | 82.7 | 76.8 | 67.0 | 79.8 | 61.0 | 66.3 |
| (+) Col loss + Real reward | 82.7 | 76.7 | 67.0 | 78.0 | 60.0 | 65.4 |
| (+) Com loss + Binary reward | 82.4 | 76.8 | 69.0 | 80.9 | 59.7 | 65.4 |
| (+) Com loss + Real reward | 82.6 | 76.8 | 68.0 | 77.6 | 60.0 | 64.4 |
| (-) Curriculum learning + Col loss | 82.5 | 75.0 | 67.0 | 78.7 | 59.4 | 63.5 |
| (-) Curriculum learning + Comp loss | 82.6 | 75.0 | 64.0 | 80.5 | 61.3 | 64.4 |

(b) DeBERTa Teacher

Table 1: Results of our method (MPDistil) on the SuperGLUE dev set with the BERT and DeBERTa teacher models. We report the accuracy scores for each of these classification tasks. We observe the teacher and student base models trained on the original train set and compare them against the distilled student and the meta-teacher model. We highlight the cases in blue in which the meta-teacher outperforms the original teacher model. Scenarios highlighted with brown are the cases in which the distilled student model performs better than the teacher.

To assess the effectiveness of MPDistil in distilling knowledge from the teacher to the student models, we compare it with other competitive knowledge distillation techniques. As the performance of the distilled student depends on the teacher's performance, we derive an evaluation metric, $\Delta\text{Margin} = (\text{Performance of student} - \text{Performance of teacher})$, to compare the distillation methods. The metric indicates the student's performance in terms of the teacher (or the improvement over the teacher), which is agnostic to the original teacher's performance. It is worth noting that existing distillation techniques can not empower the student model to perform better than the teacher model, often leading to a negative $\Delta\text{Margin}$. We report $\Delta\text{Margin}$ for different distillation techniques on the SuperGLUE dev in Table 2 (with BERT models) and Table 3 (with DeBERTa models). With the BERT teacher model, MPDistil

| Methods | BoolQ | CB | COPA | RTE | WiC | WSC |
|---|---|---|---|---|---|---|
| KD Hinton et al. (2015) | -13.3 | -19.1 | -4.3 | -3.7 | -9.1 | -14.4 |
| PD Turc et al. (2019) † | -9.6 | -9.5 | -0.3 | -13.5 | -6.9 | -11.2 |
| PKD Sun et al. (2019) | -1.7 | -5.9 | -6.0 | -3.8 | -0.4 | -12.5 |
| DistilBERT Sanh et al. (2019) † | -6.0 | -7.7 | -1.0 | -12.0 | -5.8 | -9.3 |
| Theseus Xu et al. (2020) † | -1.6 | -3.6 | -4.3 | -4.8 | -1.8 | -11.5 |
| TinyBERT Jiao et al. (2019) | **-1.4** | -1.2 | 4.3 | -3.7 | 1.7 | -2.9 |
| MobileBERT Sun et al. (2020) † | -4.8 | -2.4 | -0.7 | -14.0 | -2.3 | -9.3 |
| SID Aguilar et al. (2020) † | -10.1 | -17.3 | -1.0 | -14.8 | -9.0 | -12.8 |
| MiniLM Wang et al. (2020b) † | -3.5 | -11.9 | -4.0 | -5.3 | -1.2 | -14.4 |
| MiniLMv2 Wang et al. (2020a) † | -2.7 | -14.3 | -4.0 | -6.3 | -2.5 | -15.1 |
| ALP-KD Passban et al. (2021) † | -2.2 | -11.3 | -5.3 | -4.8 | -1.3 | -13.1 |
| LRC-BERT Fu et al. (2021) † | -4.5 | -9.5 | -0.3 | -16.4 | -8.5 | -11.2 |
| Annealing-KD Jafari et al. (2021) † | -8.8 | -5.9 | 3.3 | -14.0 | -6.3 | -11.2 |
| CKD Park et al. (2021) † | -7.8 | -6.6 | -1.0 | -11.7 | -7.3 | -11.2 |
| Universal-KD Wu et al. (2021a) † | -1.8 | -5.4 | -7.3 | -2.8 | -0.6 | -11.2 |
| DIITO Wu et al. (2021b) † | -3.9 | -5.9 | 6.0 | -7.5 | -5.4 | -8.6 |
| Continuation-KD Jafari et al. (2022) † | -8.0 | -7.1 | 2.7 | -14.2 | -7.9 | -13.1 |
| RAIL-KD Haidar et al. (2021) † | -10.4 | -7.7 | 0.7 | -12.4 | -5.8 | -7.7 |
| MGSKD Liu et al. (2022a)) † | -6.1 | -6.6 | -1.0 | -7.0 | -3.0 | -12.8 |
| MetaDistil Zhou et al. (2021) | -2.7 | -1.8 | 1.0 | -2.0 | -1.6 | 0.9 |
| MPDistil (Ours) | -1.9 | **0.0** | **7.0** | **0.4** | **2.5** | **1.0** |
| (-) Curriculum learning | -2.8 | -5.3 | -4.0 | -1.8 | 1.2 | 0.0 |

Table 2: $\Delta\text{Margin}$ reported for different distillation methods on SuperGLUE tasks dev split with BERT models. The higher the value of $\Delta\text{Margin}$, the more effective the distillation technique. $\Delta\text{Margin} > 0$ indicates that the student outperforms the teacher. The results highlighted with † are taken from Tan et al. (2023). For MPDistil and the ablation, we use the best scores obtained across different loss and reward configurations.

shows a significant $\Delta\text{Margin}$ of $+1.5\%$ on average compared to two most competitive baselines, TinyBERT (average $\Delta\text{Margin} = -0.5\%$) and MetaDistil (average $\Delta\text{Margin} = -1.0\%$). Even on the SuperGLUE test split, the BERT student model distilled with MPDistil outperforms the BERT teacher model on three out of six tasks (Table 5 in Appendix B.4). The margin between the teacher and student models is higher with the DeBERTa model. On COPA and RTE tasks, the DeBERTa-v2-xxlarge model achieves nearly state-of-the-art results, whereas the 12-layered student model achieves significantly lower performances. However, the average performance drop with MPDistil distillation is $4.6\%$, significantly lower than the other distillation methods – CKD ($9.8\%$) and MetaDistil ($6\%$). MPDistil shows similar competitive performance on GLUE dev set (Table 4). BERT-6L distilled with MPDistil achieves an average improvement of $3.1\%$ over the base student model. Similar improvements are observed on the GLUE test set (Table 7 in Appendix B.4). In terms of accuracy, the student model distilled with MPDistil outperforms the teacher BERT-base model on four GLUE tasks on both the dev and test split. Table 6 and Table 8 in Appendix B.4 show that MPDistil achieves the highest $\Delta\text{Margin}$ among all the KD techniques on four tasks on both GLUE dev and test datasets.

| Methods | MNLI/MNLI-MM | MRPC | QNLI | QQP | RTE | SST-2 | STS-B | WNLI |
|---|---|---|---|---|---|---|---|---|
| | Acc | Acc/F1 | Acc | Acc/F1 | Acc | Acc | Pear/Spear | Acc |
| Teacher (BERT-base) | 83.7/83.4 | 84.0/88.2 | 91.0 | 90.5/87.3 | 67.1 | 89.7 | 89.4/89.1 | 56.3 |
| Student (BERT-base 6L) | 77.8/78.3 | 75.8/81.5 | 85.0 | 89.1/85.0 | 64.6 | 87.7 | 88.8/88.5 | 56.3 |
| Distilled Student | 80.3/80.8 | 80.0/86.2 | 87.6 | 90.2/86.9 | 66.4 | 90.4 | 89.1/88.8 | 56.3 |
| *Meta-teacher* | | | | | | | | |
| (+) Col loss | 83.9/83.8 | 83.8/88.1 | 91.1 | 90.5/87.6 | 67.8 | 93.5 | 89.2/89.1 | 59.1 |
| (+) Com loss | 83.9/83.8 | 84.1/88.2 | 91.2 | 90.5/87.6 | 67.8 | 93.2 | 89.4/89.1 | 56.3 |
| *Student with* `MPDistil` | | | | | | | | |
| (+) Col loss + Binary reward | 80.5 /80.8 | 81.4 /86.7 | 87.6 | 90.2 /86.9 | 67.1 | 91.0 | 89.1 /88.9 | 63.3 |
| (+) Col loss + Real reward | 80.4/80.2 | 81.6/86.8 | 87.7 | 90.3/87.0 | 66.4 | 90.9 | 89.1/88.9 | 56.3 |
| (+) Com loss + Binary reward | 80.4/80.7 | 81.6/86.8 | 87.6 | 90.3/87.0 | 67.5 | 91.4 | 89.1/88.9 | 56.3 |
| (+) Com loss + Real reward | 80.3/81.0 | 81.5/86.8 | 87.7 | 90.3/87.0 | 66.4 | 90.9 | 89.1/88.9 | 56.3 |
| (-) Curriculum learning + Col loss | 79.7/81.0 | 82.0/87.2 | 88.3 | 90.2/87.0 | 65.3 | 90.8 | 89.1/88.8 | 56.3 |
| (-) Curriculum learning + Com loss | 79.8/81.0 | 81.6/87.1 | 88.3 | 90.2/86.8 | 65.3 | 90.9 | 89.1/88.8 | 56.3 |

Table 4: Results of our method (`MPDistil`) on the GLUE dev set with BERT model.

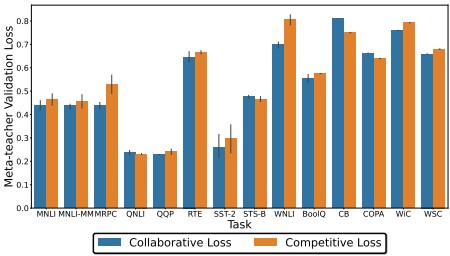

Figure 2: Meta-teacher validation loss w.r.t. different meta-teacher loss formulations.

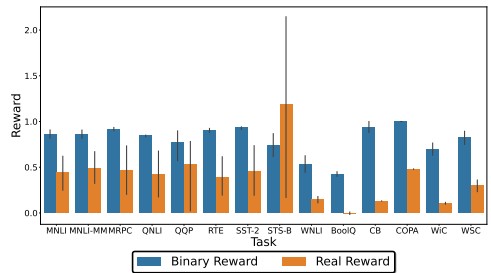

Figure 3: Student rewards for different tasks.

| Methods | BoolQ | CB | COPA | RTE | WiC | WSC |
|---|---|---|---|---|---|---|
| PKD Sun et al. (2019) | -4.7 | **-1.8** | -22.0 | -11.9 | -1.9 | 0.0 |
| CKD Park et al. (2021) | -6.8 | -7.3 | -27.0 | -13.0 | -0.2 | -4.8 |
| MetaDistil Zhou et al. (2021) | -4.6 | -8.9 | **-13.0** | -10.8 | -1.1 | 2.3 |
| `MPDistil` (Ours) | **-4.1** | -5.3 | -15.0 | **-9.7** | 3.3 | **2.9** |
| (-) Curriculum learning | -4.2 | -7.1 | -17.0 | -10.1 | **3.6** | 1.0 |

Table 3: ΔMargin for different distillation methods on the dev split of SuperGLUE tasks with DeBERTa models.

We evaluate `MPDistil` with OPT-1.3B (Zhang et al., 2022), a decoder-only LLM, reported in Appendix B.4 Table 12. The average ΔMargin on SuperGLUE tasks with MPDistil is $-0.58$, significantly lower than PKD which achieves an margin of $-1.92$(c.f. Table 13). We report the standard deviation of performances obtained in each episode of curriculum learning in Tables 9, 10 and 11 in Appendix B.4. With BERT, the average standard deviation across all SuperGLUE and GLUE tasks remain meagre at $2.4\%$ and $0.71\%$, respectively, indicating our framework's stability.

## 5 DISCUSSION

**Can better learners teach better?** Figure 2 highlights the distribution of meta-teacher validation loss, and shows lower validation loss with collaborative loss for most tasks than competitive loss. As Proposition 1 suggested, lower collaborative loss than competitive loss asserts stronger students. We analyze how a lower loss enables a better meta-teacher and encourages a better student. Towards this, we observe the improvement shown by the meta-teacher over the original fine-tuned teacher. Similarly, we calculate the improvements the distilled student shows over the fine-tuned student model. The Pearson's correlation coefficient between meta-teacher improvement and student improvement under collaborative loss is $0.65$ ($p$-value=$5e-8$); in contrast, the same under competitive loss is only $0.02$ with an insignificant $p$-value of $0.86$. Overall, the Pearson's correlation coefficient between the meta-teacher and student improvement values remains $0.40$ ($p$-value=$1e-5$). With such a low $p$-value and a strong positive correlation, we assert that a stronger meta-teacher leads to a better and more competitive student.

**How does higher student reward impact student's performance?** Figure 3 highlights the distribution of rewards gathered by the distilled student across tasks. On the classification tasks, the expected binary reward obtained is $0.81$, as opposed to the expected real reward of $0.35$. As per Proposition 2, the expected binary reward on these classification tasks is always higher than the expected real reward. With binary reward formulation, the correlation between the performances of the distilled student and the meta-teacher is $0.77$, significantly higher than the correlation with the real reward ($0.60$). On BoolQ, the expected binary reward is $< 0.5$, indicating that the student model is more underconfident than the teacher model. This propagates to the poorer dev performance by

the distilled student model. In contrast, on the tasks with the highest expected reward such as RTE, SST-2, CB and COPA, the distilled student outperforms the teacher model with the widest margins.

**How does a student select learning curricula?**  We analyze the cumulative weightage of different tasks in the student curriculum in Figure 4a.  On the low-performing tasks such as MNLI, MNLI-mm, and QNLI, the model explores all the different tasks with nearly uniform weightage.  Compared to this, on high-performing tasks such as RTE, CB, and COPA, the model perfectly balances exploration and exploitation in selecting the suitable curriculum.  For instance, on CB, the model only chooses the COPA task in its curriculum.

On the other hand, on COPA, CB has the highest weightage in the curriculum. On the RTE task, the model selects the same task repeatedly in the curriculum. To understand how the model preserves the curriculum throughout its learning, we calculate a similarity metric, $sim_{i,i+1} = \frac{|\text{LCS}(C_i, C_{i+1})|}{|C_i|}$, between two learned curricula $C_i$ and $C_{i+1}$ in two consecutive episodes. LCS denotes the longest common subsequence. We visualize the $sim_{i,i+1}$ distribution for different tasks in Figure 4b. A high similarity score indicates that the model only exploits and shows reluctance to learn new curricula. Interestingly, the model has lower average similarity on the high-performing tasks, indicating more exploration. However, after learning the most suitable curriculum, the model stops exploration and exploits the same set of tasks to preserve high rewards. On the other hand, on low-performing tasks like MNLI, MNLI-mm and QNLI, the model shows higher average similarity with low variance, indicating more restrictive behavior.

We further calculate the count of each task in each curriculum and the chi-square distance between consecutive curricula to understand how different tasks are preserved within curricula. We report the distribution of chi-square distances in Figure 4c. For high-performing tasks, we observe a significantly lower chi-square distance between consecutive curricula. Lower chi-square and low average $sim$ values conclude that for these tasks, the curriculum learning model does not explore newer tasks in the curriculum but rather explores the different sequences among them. It justifies that not only are the tasks important within curricula, but their sequence also equally matters.

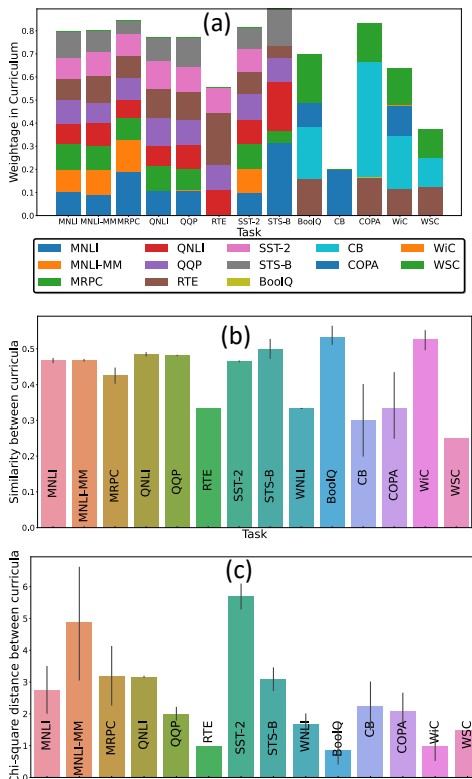

Figure 4: (a) Cumulative weightage of different tasks in final student curriculum for SuperGLUE and GLUE benchmarks. (b) Distribution of curriculum similarities for different SuperGLUE and GLUE tasks. High curriculum similarity indicates that the student model preserves the curriculum in consecutive iterations. (c) Distribution of Chi-square distance between consecutive curricula.

## 6  CONCLUSION

This paper introduced `MPDistil`, a method for distilling meta-policy knowledge, emphasizing the importance of creating a suitable joint utility to enhance the teacher's distillation capabilities. Furthermore, we underscored the necessity of employing multi-task curricula to produce more robust and effective student models through teacher knowledge distillation. It is essential to note that our framework is generic and can be applied to distilling large language models, regardless of their underlying architecture. While our empirical study primarily focused on natural language understanding tasks, our methodology can also be extended to other tasks involving natural language generation and reasoning. While our student curricula were designed with tasks featuring similar abstractions, it would be worth noting the potential impact of diverse tasks with varying levels of abstraction and complexity on enhancing the generalizability of student models.

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

## A PROOFS OF THE THEORETICAL RESULTS

### A.1 PROOF OF PROPOSITION 1

The expected collaborative and competitive loss for any given teacher and student logit is defined as $\mathbb{E}[-\log \bar{y}_{(i,T)} - \log \bar{y}_{(i,S)}]$ and $\mathbb{E}[-2\log \bar{y}_{(i,T)} + \log \bar{y}_{(i,S)}]$, respectively.

$$\mathbb{E}[\mathcal{L}_{\mathcal{T}}^{\text{meta col}}] < \mathbb{E}[\mathcal{L}_{\mathcal{T}}^{\text{meta com}}]$$
$$\iff \mathbb{E}[-\log \bar{y}_{(i,T)} - \log \bar{y}_{(i,S)}] < \mathbb{E}[-2\log \bar{y}_{(i,T)} + \log \bar{y}_{(i,S)}]$$
$$\iff \mathbb{E}[\log \bar{y}_{(i,T)}] < 2\mathbb{E}[\log \bar{y}_{(i,S)}]$$
$$\iff \mathbb{E}[\log \bar{y}_{(i,T)}] < \mathbb{E}[\log \bar{y}_{(i,S)}^2]$$
$$\iff \mathbb{E}[\log \bar{y}_{(i,S)}^2 - \log \bar{y}_{(i,T)}] > 0$$
$$\implies \log \mathbb{E}[\bar{y}_{(i,S)}^2] - \log \mathbb{E}[\bar{y}_{(i,T)}] > 0 \text{ (Jensen's inequality)}$$
$$\iff \log \frac{\mathbb{E}[\bar{y}_{(i,S)}^2]}{\mathbb{E}[\bar{y}_{(i,T)}]} > 0$$
$$\iff \frac{\mathbb{E}[\bar{y}_{(i,S)}^2]}{\mathbb{E}[\bar{y}_{(i,T)}]} > 1.$$

As $\bar{y}_{(i,S)} > \bar{y}_{(i,S)}^2$, it proves that $\mathbb{E}[\bar{y}_{(i,S)}] > \mathbb{E}[\bar{y}_{(i,T)}]$. ∎

## A.2 PROOF OF PROPOSITION 2

$$\mathbb{E}[\mathbb{1}_{\hat{y}_{(i,S)}>\hat{y}_{(i,T')}}] = \mathbb{P}_{\hat{y}_{(i,S)},\hat{y}_{(i,T')}}[\hat{y}_{(i,S)} - \hat{y}_{(i,T')} > 0]$$
$$= \int_{\hat{y}_{(i,S)}-\hat{y}_{(i,T')}=0}^{1} \int_{\hat{y}_{(i,T')}=0}^{1} d\hat{y}_{(i,S)} \cdot d\hat{y}_{(i,T')}.$$

Being two probability distributions, the random variable $\hat{y}_{(i,S)} - \hat{y}_{(i,T')}$ has its range $[-1, 1]$.

$$\mathbb{E}[\hat{y}_{(i,S)} - \hat{y}_{(i,T')}] = \int_{\hat{y}_{(i,S)}=0}^{1} \int_{\hat{y}_{(i,T')}=0}^{1} \left( \hat{y}_{(i,S)} - \hat{y}_{(i,T')} \right) \cdot d\hat{y}_{(i,S)} \cdot d\hat{y}_{(i,T')}$$
$$= \int_{\hat{y}_{(i,S)}=\hat{y}_{(i,T')}}^{1} \int_{\hat{y}_{(i,T')}=0}^{1} \left( \hat{y}_{(i,S)} - \hat{y}_{(i,T')} \right) \cdot d\hat{y}_{(i,S)} \cdot d\hat{y}_{(i,T')}$$
$$+ \int_{\hat{y}_{(i,S)}=0}^{\hat{y}_{(i,T')}} \int_{\hat{y}_{(i,T')}=0}^{1} \left( \hat{y}_{(i,S)} - \hat{y}_{(i,T')} \right) \cdot d\hat{y}_{(i,S)} \cdot d\hat{y}_{(i,T')}.$$

The second term in the above expression is less than or equal to 0 as $\hat{y}_{(i,S)} - \hat{y}_{(i,T')} \leq 0$ for any $\hat{y}_{(i,S)} \in [0, \hat{y}_{(i,T')}]$. Therefore,

$$\mathbb{E}[\hat{y}_{(i,S)} - \hat{y}_{(i,T')}] \leq \int_{\hat{y}_{(i,S)}=\hat{y}_{(i,T')}}^{1} \int_{\hat{y}_{(i,T')}=0}^{1} \left( \hat{y}_{(i,S)} - \hat{y}_{(i,T')} \right) \cdot d\hat{y}_{(i,S)} \cdot d\hat{y}_{(i,T')}.$$
$$= \int_{\hat{y}_{(i,S)}-\hat{y}_{(i,T')}=0}^{1-\hat{y}_{(i,T')}} \int_{\hat{y}_{(i,T')}=0}^{1} d\hat{y}_{(i,S)} \cdot d\hat{y}_{(i,T')} \text{ (Change of limits)}$$
$$\leq \int_{\hat{y}_{(i,S)}-\hat{y}_{(i,T')}=0}^{1} \int_{\hat{y}_{(i,T')}=0}^{1} d\hat{y}_{(i,S)} \cdot d\hat{y}_{(i,T')}$$
$$= \mathbb{E}[\mathbb{1}_{\hat{y}_{(i,S)}>\hat{y}_{(i,T')}}]. \quad ∎$$

# B EXPERIMENTAL DETAILS AND RESULTS

## B.1 DATASETS

This section briefly describes the tasks of SuperGLUE (Wang et al., 2019) and GLUE (Wang et al., 2018) benchmarks.

### B.1.1 SuperGLUE Benchmark

**BoolQ:** Boolean Questions comprise of binary questions using the Google search engine as their source of questions; they are then paired with appropriate paragraphs from Wikipedia articles that contain the relevant answers (Clark et al., 2019).

**CB:** CommitmentBank comprises of short texts with embedded clauses. The examples are taken from sources like British National Corpus Fiction and Wall Street Journal. It involves a three-class textual entailment task. Each example includes a premise and the corresponding hypothesis along with the class label "contradiction", "neutral", or "entailment" (De Marneffe et al., 2019).

**COPA:** Choice of Plausible Alternatives is a causal reasoning task which involves selecting the most plausible choice for a cause or effect given a premise. (Roemmele et al., 2011).

**WiC:** Word-in-Context is a task focused on word sense disambiguation, comprising of binary classification of pairs of sentences. In this task, two text snippets are provided, each containing a word that could have multiple meanings. The goal is to ascertain whether the word has the same meaning in both sentences (Pilehvar & Camacho-Collados, 2018).

**WSC:** Winograd Schema Challenge is a coreference resolution task where each sentence example includes a passage, a pronoun, and a choice of nouns from that sentence. The task objective is to determine whether the pronoun and noun refer to each other or not (Levesque et al., 2012).

### B.1.2 GLUE Benchmark

**SST-2:** Stanford Sentiment Treebank is a dataset derived from movie reviews, annotated with sentiment labels. The goal is to predict the sentiment ("positive"/"negative") of the reviews, employing a two-way classification Socher et al. (2013).

**MRPC:** Microsoft Research Paraphrase Corpus consists of sentence pairs obtained from online news sources and are human-annotated to ascertain their semantic equivalence. This task involves predicting whether two given sentences are paraphrases of each other or not (Dolan & Brockett, 2005).

**STS-B:** Semantic Textual Similarity Benchmark predicts the degree of similarity between two given sentences on a scale from 0-5. The dataset is compiled from diverse origins, including videos and image descriptions, news headlines, and natural language inference information. (Cer et al., 2017).

**QQP:** Quora Question Pairs involves predicting whether two given questions asked on the question-answering website Quora are semantically equivalent or not (Chen et al., 2018).

**MNLI:** Multi-Genre Natural Language Inference involves predicting whether a given hypothesis is "entailed", "neutral", or "contradicted" for a given premise. The dataset is collected from various sources, including fiction, government reports, and transcribed speech (Williams et al., 2017).

**QNLI:** Question-answering NLI is a task based on the Stanford Question Answering Dataset. It involves determining whether a given context sentence contains the answer to a corresponding question Wang et al. (2018).

**RTE:** Recognizing Textual Entailment is compiled from collection of datasets that originate from a series of annual textual entailment challenges (Dagan et al., 2005; Haim et al., 2006; Giampiccolo et al., 2007; Bentivogli et al., 2009). These datasets are constructed using news and Wikipedia text sources. The datasets are transformed into a two-class split, collapsing "neutral" and "contradiction" categories into "not entailment".

**WNLI:** Winograd NLI requires selecting the referent of a pronoun from choices by converting it into sentence pair classification. The task assesses whether substituting the pronoun with possible referents results in sentence entailment (Levesque et al., 2012).

### B.2 Experimental Setup

For all the SuperGLUE and GLUE tasks, we use a maximum sequence length of 128. All the results reported in the paper (except for the results obtained from other baselines) are obtained from running a grid search over the set of hyperparameters, teacher learning rate from {2e-5, 3e-5}, the student

learning rate from {2e-5, 3e-5}, $\tau$ from {4.0, 5.0, 6.0, 7.0}, $\alpha$ from {0.4, 0.5, 0.6}, and $\beta$ from {80, 90, 100}. The discount factor $\gamma$ is set as 0.99. The meta-teacher and the curriculum models are trained with a fixed learning rate of 0.001. We train all the models with a maximum of 10 epochs, and the curriculum model is trained with 200 episodes. We set the training, quiz and validation batch size as 8. All the models are trained with Adam optimizer with weight decay (Loshchilov & Hutter, 2017). One Tesla V100 and A100-40 GPU were used for conducting the experiments.

### B.3 BASELINES

To evaluate the impact of our proposed distillation method, we undertake several competitive KD and MetaKD baselines.

**PKD** (Sun et al., 2019) uses a patient learning approach where instead of using the last layer of teacher for distilling knowledge into the student, the student utilises the intermediate layer representation of the teacher model.

**CKD** (Park et al., 2021) transfers knowledge from teacher to student, utilising the word representations contextual knowledge. It defines two contextual knowledge objectives – *Word Relation (WR)* and *Layer Transforming Relation (LTR)*, examining the relationships among words in a particular layer and across different layers, respectively.

**TinyBERT** (Jiao et al., 2019) employs a two-stage framework, general and task-specific distillation of BERT-base into a tiny 4-layer student model, using a multiple distillation objective. In addition to logit-based matching, the teacher transfers its knowledge through embedding layers, intermediate hidden states, and attention matrices.

**MetaDistil:** Zhou et al. (2021) argued that the teacher model is not optimized for transferring knowledge to the student model. MetaDistil proposes a trainable teacher setup wherein the student's predictive performance is considered while fine-tuning the teacher model using a *quiz* dataset. The updated teacher is further used to distil knowledge into the student further.

### B.4 RESULTS

Table 5 and Table 7 highlight different model performances on SuperGLUE and GLUE test[4] sets, respectively. BERT-6L distilled with `MPDistil` achieves better accuracy than the BERT-base teacher model in three SuperGLUE and four GLUE tasks. Table 6 and Table 8 highlight the $\Delta$Margin for `MPDistil` on GLUE dev and test set respectively. It is worth noting that most contemporary KD methods fail to improve the student's performance over the teacher's on GLUE test. In contrast, the student model distilled with `MPDistil` achieves positive $\Delta$Margin on MRPC, QQP, RTE and SST-2. The ablation model without curriculum learning achieves positive $\Delta$Margin on the STS-B task. These superior performances demonstrate the generalizability of the student model distilled with `MPDistil`.

| Methods | BoolQ | CB | COPA | RTE | WiC | WSC |
|---|---|---|---|---|---|---|
| Teacher (BERT-base) | 74.6 | 83.6 | 56.6 | 63.2 | 54.8 | 65.1 |
| Student (BERT-base 6L) | 71.3 | 81.2 | 53.0 | 64.1 | 56.4 | 65.1 |
| Distilled Student | 72.1 | 82.0 | 52.8 | 64.3 | 53.9 | 65.1 |
| Meta-teacher | | | | | | |
| (+) Col loss | 73.0 | 84.0 | 56.8 | 62.8 | 54.9 | 65.1 |
| (+) Com loss | 72.9 | 84.0 | 58.8 | 62.9 | 54.4 | 64.4 |
| Student with `MPDistil` | | | | | | |
| (+) Col loss + Binary reward | 72.1 | 81.2 | 50.6 | 64.2 | 53.8 | 63.0 |
| (+) Col loss + Real reward | 71.1 | 82.4 | 53.6 | 64.2 | 53.8 | 64.4 |
| (+) Com loss + Binary reward | 71.2 | 82.0 | 51.6 | 64.5 | 55.0 | 65.8 |
| (+) Com loss + Real reward | 71.3 | 81.6 | 50.6 | 64.2 | 54.7 | 66.4 |
| (-) Curriculum learning + Col loss | 72.4 | 82.0 | 53.0 | 62.8 | 54.3 | 65.1 |
| (-) Curriculum learning + Comp loss | 72.7 | 82.4 | 53.0 | 64.2 | 55.2 | 65.8 |

Table 5: Results of our method on the SuperGLUE test sets with the BERT model.

---

[4]Submissions to `https://super.gluebenchmark.com/` and `https://gluebenchmark.com/`

| Methods | MNLI/MNLI-MM Acc | MRPC Acc/F1 | QNLI Acc | QQP Acc/F1 | RTE Acc | SST-2 Acc | STS-B Pear/Spear |
|---|---|---|---|---|---|---|---|
| KD Hinton et al. (2015) † | -2.0/-1.7 | -2.4/-2.0 | -2.0 | -0.5/-1.2 | -3.7 | -1.8 | -1.6/-1.6 |
| PKD Sun et al. (2019) † | -1.9/-1.6 | -2.9/-2.2 | -1.7 | -0.5/-0.7 | -3.8 | -1.7 | -1.6/-1.7 |
| TinyBERT Jiao et al. (2019) † | **-1.0/-1.1** | -1.1/-1.1 | -1.4 | -0.8/-0.9 | -3.7 | -1.1 | -1.0/-1.1 |
| RCO Jin et al. (2019) † | -2.2/-2.0 | -2.5/-2.1 | -1.5 | -0.8/-1.1 | -3.8 | -1.6 | -1.5/-1.5 |
| TAKD Mirzadeh et al. (2020) † | -2.1/-1.9 | -2.6/-2.0 | -1.6 | -0.7/-1.0 | -2.9 | -1.6 | -2.0/-1.8 |
| DML Zhang et al. (2018) † | -2.2/-2.0 | -2.5/-2.0 | -1.6 | -1.1/-1.1 | -3.0 | -1.5 | -1.8/-1.7 |
| ProKT Shi et al. (2020) † | -1.8/-1.7 | -1.3/-0.9 | -1.5 | -0.5/-0.6 | -3.0 | -1.7 | -1.3/-1.2 |
| SFTN Park et al. (2021) † | -2.2/-2.0 | -2.3/-1.8 | -1.7 | -1.0/-1.0 | -2.9 | -1.5 | -1.8/-1.3 |
| MetaDistil Zhou et al. (2021)† | -1.1/-1.1 | **-0.8/-0.5** | **-0.8** | -0.4/-0.4 | -2.0 | -0.7 | -0.8/-0.7 |
| MPDistil (Ours) | -3.2/-2.3 | -2.4/-1.4 | -3.3 | **-0.2/-0.3** | **0.4** | 1.1 | **-0.3/-0.2** |
| (-) Curriculum learning | -4.0/-2.4 | -2.0/-1.0 | -2.6 | 0.3/**-0.3** | -1.8 | **1.3** | -0.4/-0.3 |

Table 6: ΔMargin reported for different distillation methods on GLUE tasks dev split with the BERT model. The results highlighted with † are obtained from Zhou et al. (2021).

| Methods | MNLI/MNLI-MM Acc | MRPC Acc/F1 | QNLI Acc | QQP Acc/F1 | RTE Acc | SST-2 Acc | STS-B Pear/Spear | WNLI Acc |
|---|---|---|---|---|---|---|---|---|
| Teacher (BERT-base) | 83.5/82.8 | 81.3/86.9 | 90.2 | 87.9/70.3 | 63.2 | 91.4 | 83.4/82.1 | 65.1 |
| Student (BERT-base 6L) | 79.5/78.2 | 80.5/86.0 | 86.9 | 87.9/68.6 | 64.1 | 90.1 | 83.8/82.5 | 65.1 |
| Distilled Student | 80.0/79.6 | 79.6/85.5 | 86.7 | 88.1/69.3 | 64.3 | 89.8 | 83.1/81.8 | 65.1 |
| Meta-teacher | | | | | | | | |
| (+) Col loss | 84.1/83.0 | 81.4/86.7 | 90.3 | 87.9/70.2 | 62.8 | 92.5 | 83.3/82.0 | 65.1 |
| (+) Com loss | 84.0/83.1 | 81.6/86.8 | 90.2 | 88.2/70.4 | 62.9 | 92.4 | 83.3/82.0 | 65.1 |
| Student with MPDistil | | | | | | | | |
| (+) Col loss + Binary reward | 80.6/79.6 | 81.4/86.8 | 86.7 | 88.1/69.3 | 64.2 | 90.4 | 83.0/81.7 | 58.9 |
| (+) Col loss + Real reward | 80.2/78.5 | 81.6/86.8 | 86.8 | 88.2/69.4 | 64.2 | 91.2 | 83.2/81.9 | 65.1 |
| (+) Com loss + Binary reward | 80.5/79.6 | 81.6/86.8 | 86.9 | 88.2/69.3 | 64.5 | 90.6 | 83.2/81.9 | 65.1 |
| (+) Com loss + Real reward | 80.2/80.1 | 81.5/86.8 | 86.8 | 88.2/69.4 | 64.2 | 91.2 | 83.2/81.9 | 65.1 |
| (-) Curriculum learning + Col loss | 79.8/79.9 | 82.0/87.3 | 88.2 | 88.5/70.4 | 62.8 | 90.2 | 84.2/83.1 | 65.1 |
| (-) Curriculum learning + Com loss | 79.4/79.9 | 81.6/87.1 | 88.2 | 88.5/70.1 | 64.2 | 91.0 | 84.1/83.1 | 65.1 |

Table 7: Results of our method (MPDistil) on the GLUE test set with the BERT model.

| Methods | MNLI/MNLI-MM Acc | MRPC Acc/F1 | QNLI Acc | QQP Acc/F1 | RTE Acc | SST-2 Acc | STS-B Pear/Spear |
|---|---|---|---|---|---|---|---|
| KD Turc et al. (2019) † | -1.8/-1.2 | -3.1/-2.1 | -1.6 | -0.3/-0.8 | -1.1 | -1.7 | - |
| PKD Sun et al. (2019) † | -3.1/-2.4 | -4.9/-3.9 | -1.5 | -0.3/-0.5 | -0.9 | -1.5 | -3.7/-4.2 |
| BERT-of-Theseus Xu et al. (2020) † | -2.2/-1.3 | -1.6/-1.3 | -0.9 | 0.1/0.4 | -0.2 | -1.3 | -1.5/-1.7 |
| ProKT Shi et al. (2020) † | -1.7/-1.2 | -2.5/-1.9 | -0.8 | -0.3/-0.3 | - | -0.2 | - |
| TinyBERT Jiao et al. (2019) † | -1.6/-0.8 | -2.0/-1.0 | -0.7 | -0.6/-0.3 | 0.4 | -0.4 | -1.3/-1.2 |
| DML Zhang et al. (2018) † | -2.0/-1.8 | -3.6/-2.4 | -1.0 | -0.5/-0.5 | -0.1 | -0.8 | -1.6/-1.8 |
| RCO Jin et al. (2019) † | -2.3/-2.2 | -3.4/-2.1 | -1.2 | -0.5/-0.8 | 0.1 | -0.9 | -1.8/-1.7 |
| TAKD Mirzadeh et al. (2020) † | -2.2/-1.7 | -3.5/-2.4 | -1.1 | -0.4/-0.6 | 0.4 | -0.6 | -1.7/-1.7 |
| SFTN Park et al. (2021) † | -2.5/-2.1 | -3.6/-2.4 | -0.9 | -0.8/-1.0 | -0.1 | -0.8 | -2.0/-1.6 |
| MetaDistil Zhou et al. (2021) † | **-0.8/-0.2** | -0.1/-0.2 | **-0.3** | -0.3/-0.1 | 0.8 | **0.0** | -1.0/-0.8 |
| MPDistil (Ours) | -3.3/-2.9 | 0.3/-0.1 | -3.4 | 0.3/-0.9 | **1.3** | -0.6 | -0.4/-0.4 |
| (-) Curriculum learning | -4.1/-2.9 | **0.7/0.4** | -2.0 | **0.6/0.1** | 1.0 | -0.4 | **0.7/1.0** |

Table 8: ΔMargin reported for different distillation methods on GLUE tasks test split with the BERT model. The results highlighted with † are obtained from Zhou et al. (2021).

| Methods | BoolQ | CB | COPA | RTE | WiC | WSC |
|---|---|---|---|---|---|---|
| Student with MPDistil | | | | | | |
| (+) Col loss + Binary reward | 72.8 ± 0.35 | 83.9 ± 3.31 | 67.0 ± 4.13 | 67.1 ± 1.44 | 58.0 ± 1.05 | 65.4 ± 5.36 |
| (+) Col loss + Real reward | 73.4 ± 0.71 | 82.1 ± 1.49 | 70.0 ± 4.22 | 66.4 ± 1.44 | 58.6 ± 0.78 | 64.4 ± 4.24 |
| (+) Com loss + Binary reward | 73.0 ± 0.63 | 80.4 ± 1.55 | 62.0 ± 3.80 | 67.5 ± 1.37 | 59.6 ± 1.63 | 65.4 ± 4.12 |
| (+) Com loss + Real reward | 73.0 ± 0.34 | 78.6 ± 2.19 | 63.0 ± 4.38 | 66.4 ± 2.22 | 58.9 ± 0.99 | 65.4 ± 2.56 |

Table 9: Performance of our method (MPDistil) along with standard deviation on the SuperGLUE dev sets with the BERT model.

| Methods | BoolQ | CB | COPA | RTE | WiC | WSC |
|---|---|---|---|---|---|---|
| Student with MPDistil | | | | | | |
| (+) Col loss + Binary reward | 82.7 ± 2.86 | 76.8 ± 3.91 | 67.0 ± 2.58 | 79.8 ± 2.25 | 61.0 ± 0.96 | 66.3 ± 0.76 |
| (+) Col loss + Real reward | 82.7 ± 0.18 | 76.7 ± 1.31 | 67.0 ± 2.87 | 78.0 ± 1.08 | 60.0 ± 0.53 | 65.4 ± 4.40 |
| (+) Com loss + Binary reward | 82.4 ± 0.49 | 76.8 ± 2.26 | 69.0 ± 3.90 | 80.9 ± 2.27 | 59.7 ± 0.47 | 65.4 ± 2.94 |
| (+) Com loss + Real reward | 82.6 ± 0.22 | 76.8 ± 1.27 | 68.0 ± 3.28 | 77.6 ± 1.28 | 60.0 ± 0.45 | 64.4 ± 3.87 |

Table 10: Performance of our method (along with standard deviation) including performance variance on the SuperGLUE dev sets with the DeBERTa model.

| Methods | MNLI/MNLI-MM Acc | MRPC Acc/F1 | QNLI Acc | QQP Acc/F1 | RTE Acc | SST-2 Acc | STS-B Pear/Spear | WNLI Acc |
|---|---|---|---|---|---|---|---|---|
| Student with MPDistil | | | | | | | | |
| (+) Col loss + Binary reward | 80.5 ± 1.38/80.8 ± 0.79 | 81.4 ± 1.05/86.7 ± 1.13 | 87.6 ± 1.67 | 90.2 ± 0.28/86.9 ± 0.44 | 67.1 ± 2.69 | 91.0 ± 0.56 | 89.1 ± 0.60/88.9 ± 0.39 | 63.3 ± 1.84 |
| (+) Col loss + Real reward | 80.4 ± 1.40/80.2 ± 1.73 | 81.6 ± 0.82/86.8 ± 0.71 | 87.7 ± 1.47 | 90.3 ± 0.49/87.0 ± 1.09 | 66.4 ± 3.74 | 90.9 ± 0.42 | 89.1 ± 0.84/88.9 ± 0.72 | 56.3 ± 2.33 |
| (+) Com loss + Binary reward | 80.4 ± 1.15/80.7 ± 0.54 | 81.6 ± 1.38/86.8 ± 1.67 | 87.6 ± 0.67 | 90.3 ± 0.51/87.0 ± 1.10 | 67.5 ± 3.96 | 91.4 ± 0.89 | 89.1 ± 0.70/88.9 ± 0.70 | 56.3 ± 0.58 |
| (+) Com loss + Real reward | 80.3 ± 1.42/81.0 ± 0.61 | 81.5 ± 0.86/86.8 ± 0.98 | 87.7 ± 1.47 | 90.3 ± 0.12/87.0 ± 0.32 | 66.4 ± 3.73 | 90.9 ± 0.49 | 89.1 ± 0.70/88.9 ± 0.69 | 56.3 ± 4.62 |

Table 11: Performance of our method (MPDistil) along with standard deviation on the GLUE dev set with BERT model.

| Methods | BoolQ | CB | COPA | RTE | WiC | WSC |
|---|---|---|---|---|---|---|
| Teacher (OPT-1.3B) | 64.9 | 83.9 | 55.0 | 53.4 | 56.7 | 63.5 |
| Student (OPT 12L) | 63.4 | 75.0 | 54.0 | 53.1 | 53.8 | 63.5 |
| Distilled Student | 65.6 | 67.9 | 55.0 | 52.7 | 54.7 | 63.5 |
| Meta-teacher | | | | | | |
| (+) Col loss | 64.9 | 80.4 | 55.0 | 50.9 | 56.1 | 63.5 |
| (+) Com loss | 65.4 | 82.1 | 55.0 | 53.8 | 56.0 | 63.5 |
| Student with MPDistil | | | | | | |
| (+) Col loss + Binary reward | 65.7 | 71.4 | 55.0 | 55.6 | 54.7 | 63.5 |
| (+) Col loss + Real reward | 65.6 | 73.2 | 55.0 | 54.9 | 56.1 | 63.5 |
| (+) Com loss + Binary reward | 65.6 | 76.8 | 55.0 | 53.8 | 56.1 | 63.5 |
| (+) Com loss + Real reward | 65.6 | 73.2 | 55.0 | 53.8 | 56.7 | 63.5 |
| (-) Curriculum learning + Col loss | 65.6 | 73.2 | 55.0 | 52.7 | 54.7 | 63.5 |
| (-) Curriculum learning + Comp loss | 64.3 | 71.4 | 54.0 | 54.5 | 54.9 | 63.5 |

Table 12: Performance of our method (MPDistil) on the SuperGLUE dev set with OPT model.

| Methods | BoolQ | CB | COPA | RTE | WiC | WSC |
|---|---|---|---|---|---|---|
| PKD Sun et al. (2019) | -0.5 | -8.9 | 0.0 | 0.7 | -2.8 | 0.0 |
| MPDistil (Ours) | **0.8** | **-7.1** | 0.0 | **1.5** | **0.0** | 0.0 |
| (-) Curriculum learning | 0.7 | -10.7 | 0.0 | 1.0 | -1.8 | 0.0 |

Table 13: ΔMargin for different distillation methods on SuperGLUE dev tasks with OPT model.

