# OpenReview forum: "A Good Learner can Teach Better: Teacher-Student Collaborative Knowledge Distillation"
_ICLR.cc/2024/Conference — ICLR 2024 poster_

### Official Review · Reviewer_cXqE · 2023-10-28

**Soundness:** 2 fair
**Presentation:** 3 good
**Contribution:** 2 fair
**Rating:** 8
**Confidence:** 3

**Summary:**

This paper presents MPDistil, a meta-policy knowledge distillation framework for language models. The motivation is to tune both teacher and student models in a collaborative way. The proposed method consists of 4 steps, teacher fine-tuning, student distillation, meta-teacher learning, and student curriculum learning. The method is compared with the state-of-the-art baselines on SuperGLUE tasks and shows advantages. In several reported results, the teacher model outperforms the original teacher models, and the student model also outperforms the original teacher models. Finally, several discussions are presented with experiments.

**Strengths:**

1. The paper is easy to follow, and the methodology is well-motivated.
2. The method is compared with the state-of-the-art baselines, and the method shows advantages.
3. Ablation study is provided to help understand the contribution of each design.
4. Several interesting discussions are presented.

**Weaknesses:**

1. Given the previous work of Zhou et al. (2021), the presented method is not very novel, since the basic idea of teaching teachers is the same.
2. The performance gain could be mostly attributed to a better teaching model. It would be better to conduct some cross-verifications by using the meta-teacher obtained in the proposed method as the teachers for other models.
3. Larger language models are encouraged to be used in the experiments to better show the value of distillation.

**Questions:**

1. Have you tried using the meta-teacher as the teach in the baselines?

---

> ### Author Response · Authors · 2023-11-16
> **Author Response to Reviewer cXqE comments**
>
> We thank the reviewer for the helpful comments. We address the raised concerns.
>
> **W1. The presented method is not very novel, Zhou et al. 2021 (MetaDistil) also proposes teaching the teacher:** Although our framework was motivated by Zhou et al. 2021 [1] (MetaDistil), there are several major drawbacks of MetaDistil, which our method, MPDistil, circumvents:
>
> * In the meta-loop of MetaDistil, the teacher model is trained to minimize the gap between the teacher and the student. Therefore, the teacher is always optimized for a low-performing student with a suboptimal objective. On the contrary, MPDistil formulates a competitive learning framework wherein both teacher and student models are encouraged to outperform each other.
>
> * MetaDistil and most of the MetaKD techniques distil knowledge for different tasks in isolation, failing to leverage the commonalities across tasks. This phenomenon is counter-intuitive in real-world settings, where a learner studies other associated subjects (*e.g.,* Mathematics and Physics) for better learning. MPDistil encourages the student model to formulate suitable learning curricula using a suitable reward system, leading to more generalized and robust student performances.
>
> The empirical results highlight the effectiveness of each of these components within MPDistil and demonstrate significant performance improvement over the contemporary KD and MetaKD methods.
>
> **W2. Gain attributed to a better teaching model. Conduct cross-verification by using meta teacher obtained from the method as a teacher for other baselines:** We distil the BERT student model using PKD (Sun et al., 2019 [2]) from the meta-teacher obtained from our method. On average, the student model distilled from the meta-teacher obtains $0.2$% higher accuracy than the student model distilled from the BERT-12L teacher model. Moreover, the statistical study reported in Section 5 also confirms the positive correlation between the meta-teacher and the student's performances. However, as shown in Tables 1-4, curriculum learning can aid an additional performance boost of $2.2$%. Therefore, competitive teacher training and student curriculum learning are equally responsible for robust student performance.
>
> **W3. Distillation results using LLMs.** As suggested, we have empirically validated our method, MPDistil, with OPT-1.3B (Zhang, Roller and Goyal et al., 2022 [3]), a decoder-only LLM. We report the accuracy and $\Delta \text{Margin}$ on SuperGLUE dev tasks with OPT-1.3B model in Tables 12 and 13 of the updated paper, respectively. The average student-teacher margin remains at $-0.58$ with MPDistil, whereas the PKD baseline achieves an average margin of $-1.9$. Moreover, the student model distilled with MPDistil outperforms the teacher model on 2 out of 6 SuperGLUE tasks, indicating its effectiveness. Table 1b and Table 3 highlight the effectiveness of MPDistil with the DeBERTa-v2-xx-large (He et al., 2020 [4]) model, the largest encoder-only language model.
>
> **Q1. Have you tried using meta-teacher as the teacher in baselines?** We distil the BERT student model using PKD (Sun et al., 2019 [2]) from the meta-teacher obtained from our method. On average, the student model distilled from the meta-teacher obtains $0.2$% higher accuracy than the student model distilled from the BERT-12L teacher model. However, the performances of the PKD-distilled student still fall short of the student model distilled with MPDistil.
>
> **References**
>
> [1] Zhou, Wangchunshu, Canwen Xu, and Julian McAuley. "BERT learns to teach: Knowledge distillation with meta learning." arXiv preprint arXiv:2106.04570 (2021).
>
> [2] Sun, Siqi, Yu Cheng, Zhe Gan, and Jingjing Liu. "Patient knowledge distillation for bert model compression." arXiv preprint arXiv:1908.09355 (2019).
>
> [3] Zhang, Susan, Stephen Roller, Naman Goyal, Mikel Artetxe, Moya Chen, Shuohui Chen, Christopher Dewan et al. "Opt: Open pre-trained transformer language models." arXiv preprint arXiv:2205.01068 (2022).
>
> [4] He, Pengcheng, Xiaodong Liu, Jianfeng Gao, and Weizhu Chen. "Deberta: Decoding-enhanced bert with disentangled attention." arXiv preprint arXiv:2006.03654 (2020).

---

> > ### Comment · Reviewer_cXqE · 2023-11-22
> > **Thank you for the response**
> >
> > Thank you for taking the time to address my concern and running additional experiments. I will raise my score.

---

### Official Review · Reviewer_upPK · 2023-10-29

**Soundness:** 4 excellent
**Presentation:** 4 excellent
**Contribution:** 3 good
**Rating:** 8
**Confidence:** 4

**Summary:**

Meta-learning-based knowledge distillation methods is a family of distillation techniques in which the teacher model is taking into account the student's performance to improve its learning. This paper introduces a new meta-learning-based knowledge distillation method which:
(i) introduces a meta-teacher model with takes as input the original teacher's and student's representations and outputs improved predictions;
(ii) introduces a reinforcement learning-based “student curriculum learning" process in which the student aims to outperform the meta-teacher by training itself on a suitable set of tasks, possibly different from the task at hand.

**Strengths:**

This is a well-written paper that introduces a novel technique that often times gives large improvements in the student's performance. The authors have made an extensive experimental evaluation comparing their results with other distillation techniques to give convincing arguments about their claims.

**Weaknesses:**

— If my understanding is correct, it seems to me that the main benefits come from the  “student curriculum learning" process that exploits data points from other tasks that are in general not available to the other baselines the authors compare against and the teacher model itself. In that sense, it is not so surprising that the student model outperforms the teacher and that the improvements margins are high enough — this should probably be made a bit more explicit by the authors. (Indeed, it is known that the student-model can certainly outperform the teacher-model in scenarios it has access to more examples, see e.g., [1, 2, 3]).

— The proposed technique is somewhat involved (implementation-wise).

[1] Self-training with Noisy Student improves ImageNet classification [Xie, Luong,  Hovy, V. Le]
[2] Does Knowledge Distillation Really Work? [Stanton, Izmailov, Kirichenko, Alemi, Wilson]
[3] Weighted Distillation with Unlabeled Examples [Iliopoulos, Kontonis, Baykal, Menghani, Trinh, Vee]

**Questions:**

I don't have any questions.

---

> ### Author Response · Authors · 2023-11-16
> **Author Response to Reviewer upPK comments**
>
> We thank the reviewer for the encouraging comments. We address the raised concerns.
>
> **W1. Major benefits arise by exploiting the data points from other tasks in student curriculum learning:** We would like to clarify that the superior performance of the student model is attributed not only to the curriculum learning framework but also to the competitive joint optimization strategy. We can observe the performances demonstrated by the ablation model without curriculum learning as empirical evidence. Interestingly, the ablation model reports positive gain on four tasks, where the distilled student model outperforms the teacher model. Tables 2 and 3 highlight the competitiveness of the ablation model without curriculum learning on SuperGLUE tasks. Section 5 also highlights the results of the statistical studies performed on the meta-teacher and the student model. The study finds a strong positive correlation between the meta-teacher improvement and the student performance. Therefore, we can conclude that only the curriculum learning framework may not be sufficient for teaching a robust student unless the teacher is robust and generalized.
>
> **W2. Similarity with other works:** We would like to clarify that Xie et al., 2019 [1] and Iliopoulos et al., 2022 [3] explore semi-supervised methodologies for better and generalized knowledge distillation to smaller student models. They leverage larger teacher models for generating synthetic data for training the student model. On the other hand, Stanton et al., 2021 [2] explore the effectiveness of knowledge distillation from the viewpoints of generalization and fidelity. Contrarily, our work explores the connection between a teacher model's learning and teaching ability. Towards this, we formulate a competitive joint learning framework, wherein the teacher and student try to outsmart each other. By doing so, both models improve their respective performances. Based on these arguments, we argue that the motivation laid by our work significantly differs from the mentioned studies.
>
> **References**
>
> [1] Xie, Qizhe, Minh-Thang Luong, Eduard Hovy, and Quoc V. Le. "Self-training with noisy student improves imagenet classification." In Proceedings of the IEEE/CVF conference on computer vision and pattern recognition, pp. 10687-10698. 2020.
>
> [2] Stanton, Samuel, Pavel Izmailov, Polina Kirichenko, Alexander A. Alemi, and Andrew G. Wilson. "Does knowledge distillation really work?." Advances in Neural Information Processing Systems 34 (2021): 6906-6919.
>
> [3] Iliopoulos, Fotis, Vasilis Kontonis, Cenk Baykal, Gaurav Menghani, Khoa Trinh, and Erik Vee. "Weighted distillation with unlabeled examples." Advances in Neural Information Processing Systems 35 (2022): 7024-7037.

---

> > ### Comment · Reviewer_upPK · 2023-11-19
> > **Reply**
> >
> > Thank you for your clarifications!

---

### Official Review · Reviewer_JBkZ · 2023-11-01

**Soundness:** 2 fair
**Presentation:** 2 fair
**Contribution:** 2 fair
**Rating:** 5
**Confidence:** 4

**Summary:**

This paper studies the problem of meta-learning based knowledge distillation, where the teacher model should be tuned during distillation.
The proposed made several adjustment to this setting with teacher fine-tuning, meta-teacher learning and student curriculum learning, each of which can somehow improve the performance.
The authors conduct extensive experiments on the natural language understanding tasks and the results somehow demonstrate the effectiveness of the proposed method.

**Strengths:**

* The meta-learning-based knowledge distillation task is interesting and still under-explored.
* The experiments are extensive.

**Weaknesses:**

* My major concern is that the paper is overall hard to follow. Specifically, in the introduction section, the authors list several challenges of MetaKD, while these challenges seems scattered and the major challenges are missed.
* The proposed method consists of several steps and the authors make several adjustment in each step. However, these adjustment seems quite straightforward, I wonder the purpose of these design and why it works.
* Concerning the detailed design of the proposed method, the meta-teacher takes hidden state representation from both the teacher and student models to generate the final output, this could be a indirect learning from ground truth labels acoress the student models. Besides the efficiency advantages, I wonder whether it is more effective to directly use the parameter-efficient fine-tuning.
* The curriculum learning has been widely studied in the literature of KD and I wonder whether it is necessary to include it in the meta-learning-based KD. The authors should provide detailed explaination on this with more ablation studies.

**Questions:**

* Please clearly organize the challenges and motivation of this paper.
* Concerning the detailed design of each part, what are they purposed for?

---

> ### Author Response · Authors · 2023-11-16
> **Author Response to Reviewer JBkZ comments**
>
> We thank the reviewer for the valuable feedback. We address the raised concerns.
>
> **W1. Drawbacks in existing meta-KD techniques:** We are sorry for the confusion. Let us elaborate on the major drawbacks of the existing meta-knowledge distillation techniques -
>
> * The major drawback of the existing MetaKD approaches is the optimization objective. In the meta-loop, the teacher model is usually trained to minimize the gap between the teacher and the student. Therefore, the teacher is always optimized for a low-performing student with a sub-optimal objective.
>
> * Majority of the MetaKD techniques distil knowledge for different tasks in isolation, failing to leverage the commonalities across tasks. This phenomenon is counter-intuitive in real-world settings, where a learner studies other associated subjects (*e.g.,* Mathematics and Physics) for better learning.
>
> **W2. Purpose of various steps in our method:** Steps 3 and 4 of our proposed framework tackle the drawbacks, as mentioned in response to W1. In step 3 of our method, we formulate collaborative and competitive loss functions that encourage the meta-teacher model to perform better than the student. In step 4, we create a reward system that encourages the student to adopt suitable optimal curricula (sequences of tasks), helping to obtain better results than the teacher. Therefore, through this competitive framework, both teacher and student learn better.
>
> **W3. Effectiveness of parameter-efficient fine-tuning:** As discussed in Section 3 of the paper, the primary objective of our designed meta-teacher is to act as a discriminator responsible for generating final class output irrespective of the representation learner. Under the competitive objective formulation, the meta-teacher aims to generate more correct labels for teacher-generated representations. Under the collaborative loss formulation, however, the meta-teacher jointly optimizes both teacher and student. Being a discriminator model, the meta-teacher does not learn the latent representations from original inputs but only consumes the representations obtained from other representation learning models. Therefore, we argue that parameter-efficient fine-tuning of the original teacher model can not discriminate the teacher and student models fairly and may be ineffective in a competitive setup.
>
> **W4. Necessity of curriculum learning in MPDistil:** Tables 1-4 contain the ablation model without curriculum learning. As elaborated in the main paper, MPDistil with curriculum learning outperforms the ablation model in 11 out of 13 tasks, with a significant margin of $2.2$%. The empirical evidence highlights the necessity of curriculum learning in meta-knowledge distillation.
>
> **Q1. Challenges and motivation of the paper:** The major challenges with the existing meta-knowledge distillation techniques can be summarized as follows -
>
> * The major drawback of the existing MetaKD approaches is the optimization objective. In the meta-loop, the teacher model is usually trained to minimize the gap between the teacher and the student. Therefore, the teacher is always optimized for a low-performing student with a suboptimal objective.
>
> * Majority of the MetaKD techniques distil knowledge for different tasks in isolation, failing to leverage the commonalities across tasks. This phenomenon is counter-intuitive in real-world settings, where a learner studies different associated subjects (*e.g.,* Mathematics and Physics) for better learning.
>
> **Q2. Purpose of step-3 and step-4 in the proposed methodology:** Steps 3 and 4 of our proposed framework tackle most of the drawbacks of the existing meta-knowledge distillation techniques. In step 3, we formulate collaborative and competitive loss functions that encourage the meta-teacher model to perform better than the student. In step 4, we create a reward system that encourages the student to adopt suitable optimal curricula (sequences of tasks), helping to obtain better results than the teacher. Therefore, through this competitive framework, both teacher and student learn better. To the best of our knowledge, the idea of training the teacher "better" has not been explored yet. In a real-world scenario, the teacher model might not always try to match the student's level for better knowledge transfer but can also try improving itself and the student model together and can even try improving by maximizing the margin from the student.

---

> > ### Author Response · Authors · 2023-11-23
> > **Kindly review our responses**
> >
> > Dear Reviewer JBkZ,
> > The discussion period is almost over. Could you please check our responses and let us know if they address your concerns? Thanks

---

### Official Review · Reviewer_AnAG · 2023-11-02

**Soundness:** 3 good
**Presentation:** 3 good
**Contribution:** 2 fair
**Rating:** 6
**Confidence:** 1

**Summary:**

This paper proposes a meta-policy distillation method MPDistill which consists of four steps. First, the teacher is fine-tuned on the task. second, the student is fine-tuned by using the task loss and the distillation loss. Third, the meta-teacher is trained to optimize the teacher's outputs and the student's representations. Fourth, a student curriculum model is trained to generate a sequence of tasks for student fine-tuning and rewards calculation. Experiments show promising results.

**Strengths:**

1. Using a curriculum model to learn a set of tasks for the student is interesting.
2. Comparison with different methods are reported.

**Weaknesses:**

1. The framework is very heavy since it consists of bi-level optimization and reinforcement learning loss.
2. The framework consists of four steps. Some important training details are missing. It is hard to reproduce.

**Questions:**

1. Meta-teacher uses a bi-optimization, which is computationally expensive. What is the training cost of the proposed method compared to the baselines?
2. The curriculum model parameters are learned via a reinforcement learning loss, which may lead to unstable training. The performance variance of the proposed method is not reported.

-----------------------------------------------------
Thanks for the authors' response. I keep my rating.

---

> ### Author Response · Authors · 2023-11-16
> **Author Response to Reviewer AnAG comments**
>
> We thank the reviewer for the valuable feedback. We address the raised concerns.
>
> **W1. The framework is very heavy:** We would like to clarify that our proposed method is one of the most optimized generalized meta-knowledge distillation methods. Traditional meta-KD techniques (e.g., MetaDistil proposed by Zhou et al., 2021 [1] ) follow a bi-level optimization to fine-tune the teacher model in a meta loop. Section 3 of our paper highlights that our formulated meta teacher has only $\sim 0.001$% parameters than the original teacher model. This makes our framework adaptable, even for large language models. Secondly, in MetaDistil, the inner-learner model is trained on the training split, whereas, in our framework, we fine-tune the student curriculum model on a much smaller (only $10$% of the original training data) quiz dataset. Therefore, in a comparable setting, the total number of gradient updates is almost $90$% less than that of MetaDistil.
>
> **W2. Important training details are missing:** We have furnished all the necessary details in Appendix B.2 to reproduce the results reported in the paper. We request the reviewer to elaborate on specific details which are missing.
>
> **Q1. Training cost of our method compared to the baselines.** In MetaDistil (one of the best Meta knowledge distillation baselines), the inner-learner model is trained on the training split, whereas, in our framework, we fine-tune the student curriculum model on a much smaller quiz dataset (only $10$% of the original training data). Therefore, in a comparable setting, the total number of gradient updates is almost $90$% less than that of MetaDistil. Moreover, MetaDistil fine-tunes the entire teacher model in the inner loop, making it unsuitable for distilling large language models. Contrarily, MPDistil uses only a discriminator meta-teacher without any language model backbone. The number of trainable parameters of our meta-teacher is $\sim 0.001$% of the original teacher model. Therefore, the meta-loop for our distillation method is significantly faster than that of MetaDistil.
>
> **Q2. Unstable curriculum learning:** As per the suggestion, we have added the standard deviation of performance metrics obtained in different episodes of curriculum learning in Appendix Tables 9-11 of the updated paper. We observe a meagre variance across different episodes for all the tasks for both BERT and DeBERTa models. For instance, On the WiC task, MPDistil shows an average standard deviation of $1.1$% with the BERT model.
>
> **References**
>
> [1] Zhou, Wangchunshu, Canwen Xu, and Julian McAuley. "BERT learns to teach: Knowledge distillation with meta learning." arXiv preprint arXiv:2106.04570 (2021).

---

> > ### Author Response · Authors · 2023-11-23
> > **Kindly review our responses**
> >
> > Dear Reviewer AnAG,
> > The discussion period is almost over. Could you please check our responses and let us know if they address your concerns? Thanks

---

### Author Response · Authors · 2023-11-20
**Response to Reviewer Comments**

Dear reviewers,

Thank you for your valuable feedback. We have addressed all the comments you have raised. Please review our rebuttals and let us know if you have further suggestions/comments/recommendations for experiments. We will be glad to address those.

---

### Author Response · Authors · 2023-11-22
**A gentle reminder to review our responses**

Dear reviewers,
The author-reviewer discussion phase will end soon (November 22nd AOE). We have approximately 24 hours remaining to receive your feedback on our responses as well as to further respond to your queries (if any). Please check our responses and let us know if they address your concerns or if further clarification is needed. We look forward to the discussion.

Thanks

---

### Meta-Review · Area_Chair_dmEC · 2023-12-05

**Metareview:**

The paper proposes MPDistill, a new knowledge distillation technique for language models. The technique consists of 4 steps
- Training the teacher model
- Standard distillation into a student model
- Meta-teacher learning
- Student curriculum learning

In the third step, the meta-teacher is trained to solve the target tasks using combined representations from the teacher and the student. There is also an additional competitive loss which encourages the meta-teacher to outperform the student.

In the last stage, the authors perform curriculum learning, using a new curreculum model trained with reinforcement learning to find an optimal curriculum to update the student.

## Strengths

- The paper reports exhaustive positive results
- Novel interesting method
- The paper addresses several limitations of prior work on meta KD

## Weaknesses

- The proposed method is quite involved, using several additional models
- Some reviewers found the presentation hard to follow
- It may be hard to attribute the success of the method to specific components

**Justification For Why Not Higher Score:**

- The proposed method is quite complicated with many moving parts
- Reviewers flagged that the paper and the method details are hard to follow

**Justification For Why Not Lower Score:**

- Reviewers were supportive of acceptance
- The authors propose a novel method with interesting results
- The method addresses limitations of prior work

---

### Decision · Program_Chairs · 2024-01-16

Accept (poster)